# Decomposition of the mean absolute error (MAE) into systematic and unsystematic components

**Scott M. Robeson**[1]*, **Cort J. Willmott**[2]

**1** Department of Geography, Indiana University, Bloomington, Indiana, United States of America,
**2** Department of Geography, University of Delaware, Newark, Delaware, United States of America

\* srobeson@indiana.edu

## Abstract

When evaluating the performance of quantitative models, dimensioned errors often are characterized by sums-of-squares measures such as the mean squared error (MSE) or its square root, the root mean squared error (RMSE). In terms of quantifying average error, however, absolute-value-based measures such as the mean absolute error (MAE) are more interpretable than MSE or RMSE. Part of that historical preference for sums-of-squares measures is that they are mathematically amenable to decomposition and one can then form ratios, such as those based on separating MSE into its systematic and unsystematic components. Here, we develop and illustrate a decomposition of MAE into three useful sub-measures: (1) bias error, (2) proportionality error, and (3) unsystematic error. This three-part decomposition of MAE is preferable to comparable decompositions of MSE because it provides more straightforward information on the nature of the model-error distribution. We illustrate the properties of our new three-part decomposition using a long-term reconstruction of streamflow for the Upper Colorado River.

**Data Availability Statement:** The data for the original reconstruction as well as the bias-corrected version used here are available from NOAA's Paleoclimatology Data site: https://www.ncei.noaa.gov/access/paleo-search/study/28810.

## Introduction

Across the sciences, model-estimation and -prediction errors are often summarized and analyzed using dimensioned [1] and dimensionless [2] measures. While dimensionless error measures have received considerable attention [3–5], dimensioned measures are better suited to summarizing the magnitude of model error in meaningful units. When given a set of model predictions ($P_i$, $i = 1, 2, \ldots, n$), where each $P_i$ corresponds to a reliable observation ($O_i$), the mean squared error (MSE) and the root mean squared error (RMSE):

$$\text{MSE} = \frac{1}{n}\sum_{i=1}^{n}(P_i - O_i)^2 \qquad (1)$$

**Funding:** The author(s) received no specific funding for this work.

**Competing interests:** The authors have declared that no competing interests exist.

$$\text{RMSE} = \left[\frac{1}{n}\sum_{i=1}^{n}(P_i - O_i)^2\right]^{\frac{1}{2}} \tag{2}$$

are routinely reported [6]. The mean absolute error (MAE):

$$\text{MAE} = \frac{1}{n}\sum_{i=1}^{n}|P_i - O_i| \tag{3}$$

is reported less often, even though it has a clearer interpretation than RMSE because MAE is the average error [7].

We and others have shown elsewhere that error statistics based on sums-of-squares have a number of issues that make them less interpretable than those based on absolute values [1, 4, 7–12]. This is especially the case when they are used as measures of average model error. An additional drawback to using MSE is that its squared dimensional units are difficult to interpret. As a result, MAE is the preferred measure of average model error. Even so, sums-of-squares measures continue to be assessed and reported, partially due to inertia, but also to their amenability to mathematical decomposition into additive variance-based measures. In the context of evaluating model error, this property was used by Willmott in 1981 [13] to decompose MSE into systematic ($\text{MSE}_s$) and unsystematic ($\text{MSE}_u$) components:

$$\text{MSE}_s = \frac{1}{n}\sum_{i=1}^{n}(\hat{P}_i - O_i)^2 \tag{4}$$

$$\text{MSE}_u = \frac{1}{n}\sum_{i=1}^{n}(P_i - \hat{P}_i)^2 \tag{5}$$

such that

$$\text{MSE} = \text{MSE}_s + \text{MSE}_u. \tag{6}$$

For both ($\text{MSE}_s$) and ($\text{MSE}_u$), ordinary least-squares (OLS) regression of the model predictions on the observations typically is used to obtain $\hat{P}_i$ (i.e., a linear fit of $P$ on $O$).

As computed above, ($\text{MSE}_s$) is typically interpreted as consistent over- and/or under-prediction of the observations by the model (i.e., the model has non-zero mean bias and/or the regression slope is not one). The unsystematic component provides an estimate of the model's random error or scatter about the regression line. Forming the ratios of $\text{MSE}_s$/MSE and $\text{MSE}_u$/MSE gives estimates of the fraction of total error (as estimated by MSE) that is identified as systematic or unsystematic. This decomposition has served as a relatively insightful summary of model error (e.g., [14]) and has been used as a guide to model improvement because a model that has a large amount of systematic error usually can be respecified to reduce the consistent over- or under-prediction.

While decomposing MSE into its constituent components has been a useful approach, MSE is a flawed measure of *average* model error. Using MSE to identify systematic and unsystmatic components of error, therefore, can produce misleading summaries of the types of errors that various models contain. Even more importantly, models may be inappropriately adjusted to reduce systematic error that has been misidentified by the MSE-based approach, e.g., when the impacts of outliers are overemphasized. As a result, our goal here is to develop and present a

more rational approach for error decomposition that uses MAE as the baseline for average model error.

## Decomposition of MAE into three components

Although our goal is to partition MAE into components that represent systematic and unsystematic error patterns, we also want to move beyond the traditional two-part decomposition to further divide systematic errors into two separate components. One can be used to indicate the amount of bias in a model and the other to represent the extent to which the model predictions systematically under- or over-estimate observations falling below and above the observed mean (the regression slope is not one). The latter is referred to as proportionality error and is distinct from model bias represented in the under- or overestimation of the observed mean.

For each of these three types of error—bias, proportionality, and unsystematic—we develop a weighting function that can be used to partition MAE into its three components. We offer a diagram (Fig 1) using a small, synthetic dataset (Table 1) to illustrate the estimation of the three components.

### Bias error

Here, we define bias as the component of systematic error that is contained in the over- or under-prediction of the observed mean. This is often referred to as the mean bias error (MBE):

$$\text{MBE} = \bar{P} - \bar{O} \tag{7}$$

In addition to indicating average over- or under-prediction, MBE can be used to develop a corresponding (to $P_i$) set of unbiased predicted values:

$$P'_i = P_i - \text{MBE} \tag{8}$$

The magnitude (absolute value) of MBE can additionally serve as the weight that determines the relative importance of bias to the overall MAE:

$$b = |\text{MBE}| . \tag{9}$$

The magnitude of bias for our example dataset (Table 1) can be seen in the bottom left panel in Fig 1, where the model predictions systematically underestimate the mean of the observations by 1.

### Proportionality error

In addition to bias error (caused by an incorrect estimation of the observed mean), there is another systematic error related to consistent under- or over-prediction. This error is reflected in the slope of the regression estimate of $P'_i$ on $O_i$ (note that if the regressions are estimated using OLS, then this slope estimate is the same as that of $P$ on $O$). If the slope of this relationship is anything other than unity, there is *proportionality* error in the model predictions. A slope of less than one indicates that the model systematically overestimates values below $\bar{O}$ and underestimates those above. Conversely, a slope greater than one indicates that the model systematically underestimates values below $\bar{O}$ and overestimates those above. To estimate proportionality error, we use the unbiased predicted regression values:

$$\hat{P}'_i = \hat{P}_i - \text{MBE} \tag{10}$$

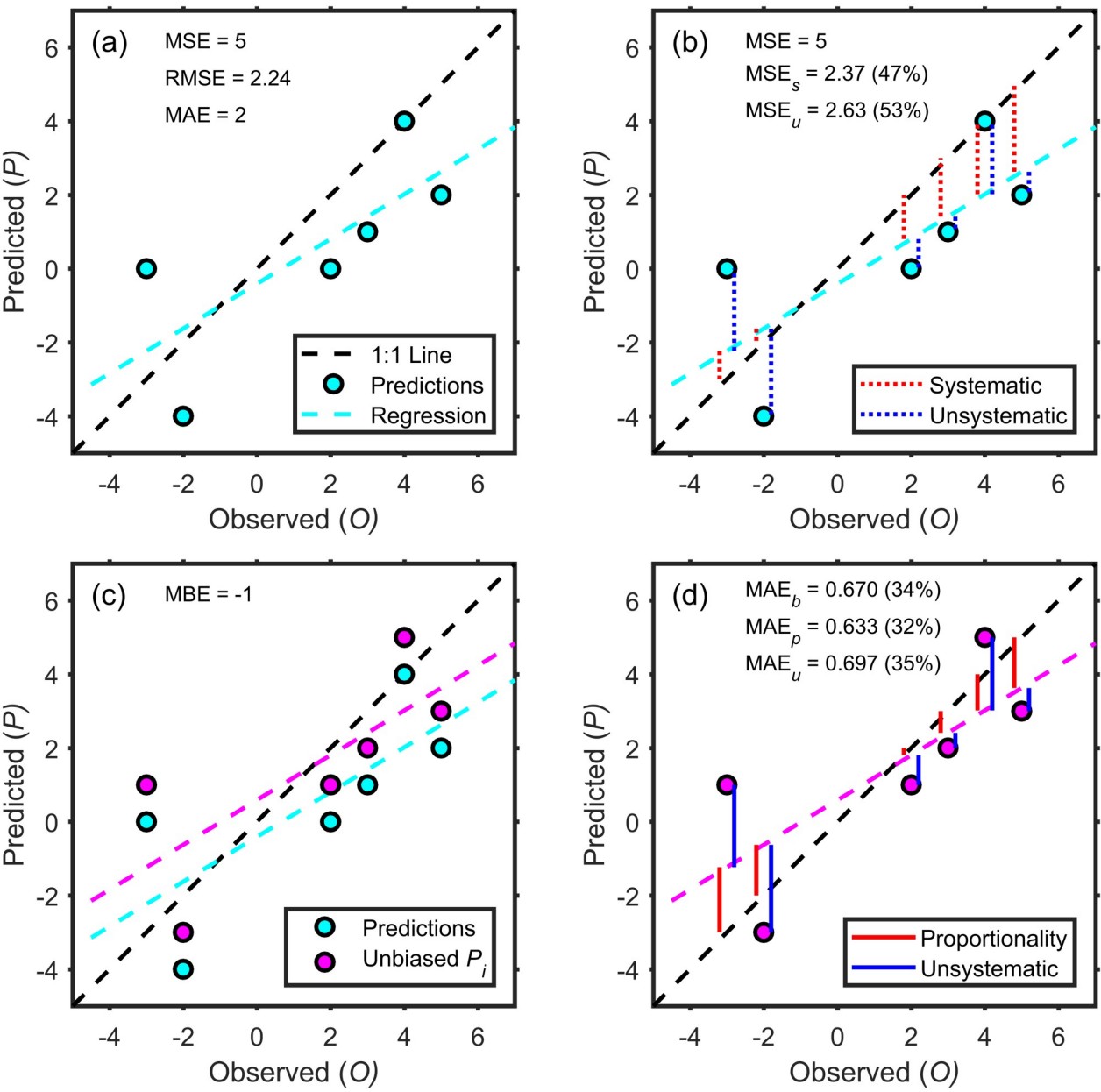

**Fig 1. Representations of model-prediction errors showing aspects of the decomposition into systematic and unsystematic components (using data from Table 1).** Predictions (in the upper left panel) can be decomposed in the traditional way using MSE, as shown in the upper-right panel where the lengths of the red and blue dotted vertical lines determine the partitioning of the errors. The bottom-left panel shows the predictions after bias is removed (i.e., $P'_i$), while the bottom-right panel shows the magnitudes of the bias, proportionality, and unsystematic components of our three-part, weight-based decomposition of MAE.

Given that the OLS solution for $\hat{P}$ is constrained to pass through $(\bar{O}, \bar{P})$, $\hat{P}'$ passes through $(\bar{O}, \bar{O})$ and, therefore, is unbiased. Weights for the relative importance of proportionality error (for each $O_i$) are determined using the difference between the unbiased predictions and the observations (the red lines in Fig 1d):

$$p_i = |\hat{P}'_i - O_i| \, . \tag{11}$$

**Table 1. Data and error components for the example in Fig 1.** For the components whose sum ($\Sigma$) is given in the far right column, dividing by 6 gives the mean value (i.e., $\bar{O}$, $\bar{P}$, MSE, MSE$_s$, (MSE$_u$), and MAE). For the other three rows that have absolute values and are used to form the $b$, $p_i$, and $u_i$ weights, which then determine MAE$_b$, MAE$_p$ and MAE$_u$ (see Eqs 14–16), the sums are not relevant and, therefore, are not given.

| Component | 1 | 2 | 3 | 4 | 5 | 6 | $\Sigma$ |
|---|---|---|---|---|---|---|---|
| Observed ($O_i$) | -3 | -2 | 2 | 3 | 4 | 5 | 9 |
| Predicted ($P_i$) | 0 | -4 | 0 | 1 | 4 | 2 | 3 |
| $(P_i - O_i)^2$ | 9 | 4 | 4 | 4 | 0 | 9 | 30 |
| $(\hat{P}_i - O_i)^2$ | 0.587 | 0.140 | 1.431 | 2.524 | 3.926 | 5.635 | 14.24 |
| $(P_i - \hat{P}_i)^2$ | 4.989 | 5.635 | 0.646 | 0.169 | 3.926 | 0.392 | 15.76 |
| $|P_i - O_i|$ | 3 | 2 | 2 | 2 | 0 | 3 | 12 |
| $b = |MBE|$ | 1 | 1 | 1 | 1 | 1 | 1 | – |
| $p_i = |\hat{P}'_i - O_i|$ | 1.766 | 1.374 | 0.196 | 0.589 | 0.981 | 1.374 | – |
| $u_i = |P'_i - \hat{P}'_i|$ | 2.237 | 2.374 | 0.804 | 0.411 | 1.981 | 0.626 | – |

## Unsystematic error

After accounting for bias and proportionality errors, the remaining error is related to scatter about $\hat{P}'$. Analogous to the way that the individual components of MSE$_u$ are formed, weights for the relative importance of each prediction's unsystematic error are determined using the difference between the unbiased predictions and the unbiased regression values:

$$u_i = |P'_i - \hat{P}'_i| \ . \tag{12}$$

Once again, if OLS regression is used for $\hat{P}'_i$, then the biased predictions and regression values produce the same weights:

$$u_i = |P'_i - \hat{P}'_i| = |P_i - \hat{P}_i| \ . \tag{13}$$

## Three-component decomposition of MAE

The three weights for bias, proportionality, and unsystematic error developed above now can be used to scale the individual components of absolute error:

$$\text{MAE}_b = \frac{1}{n} \sum_{i=1}^{n} \frac{b}{b + p_i + u_i} |P_i - O_i| \ , \tag{14}$$

$$\text{MAE}_p = \frac{1}{n} \sum_{i=1}^{n} \frac{p_i}{b + p_i + u_i} |P_i - O_i| \ , \tag{15}$$

$$\text{MAE}_u = \frac{1}{n} \sum_{i=1}^{n} \frac{u_i}{b + p_i + u_i} |P_i - O_i| \ . \tag{16}$$

A clear advantage of this weight-based decomposition of average error is that it uses MAE rather than MSE as the baseline. Another advantage is that predictions that have no error do not contribute to the components. This was not the case with the MSE-based decomposition, where predictions that have no error can substantially influence the values of MSE$_s$ and MSE$_u$ (e.g., the point that lies directly on the 1:1 line in the upper right panel of Fig 1 contributes substantially to both MSE$_s$ and MSE$_u$ despite being error-free).

It is possible for the denominator within these summations (i.e., $b + s_i + u_i$) to be zero, but that can only occur when a model has no bias and the regression line passes through a predicted value that has no error (i.e., when $b = 0$ and $P_i = \hat{P}_i = O_i$). If that rare model-prediction event occurs, those elements with $b + p_i + u_i = 0$ can simply be excluded from the summation.

Given the definitions in Eqs 14–16, $MAE_b$, $MAE_p$, and $MAE_u$ sum to MAE:

$$MAE = MAE_b + MAE_p + MAE_u \,. \tag{17}$$

As with $MSE_s$ and $MSE_u$, it is instructive to form ratios (i.e., $MAE_b/MAE$, $MAE_p/MAE$, and $MAE_u/MAE$) to identify the proportion of total error contributed by each component. The constraints within the weighted decomposition of MAE diminish $MAE_b$ relative to the magnitude of MBE. MBE, therefore, remains a useful metric to be reported when analyzing model error. R and Matlab functions for these calculations are provided in the S1 File.

## An example of model-estimation errors

To illustrate the properties of our newly derived measures of model error, we use a tree-ring based reconstruction model developed by Meko et al. [15]. This reconstruction provides over 1200 years of annually resolved flow predictions for the Upper Colorado River at Lee's Ferry. The large majority of the annual flow in the Colorado River comes from upstream of Lee's Ferry [16], so the reconstruction is an essential indicator of historical water availability for the Colorado River. The observed and reconstructed flow are water-year totals for a large river and, therefore, are reported in billions of cubic meters per year. Both the observed data and reconstructed values are based on estimates of "naturalized" streamflow, which corrects for the anthropogenic alterations of flow (i.e., reservoirs, irrigation, etc.). In a recent article [14], the model was bias-corrected so that its empirical probability distribution better matched that of the observations (e.g., compare Fig 2a and 2b). Here, we employ our new decomposition of

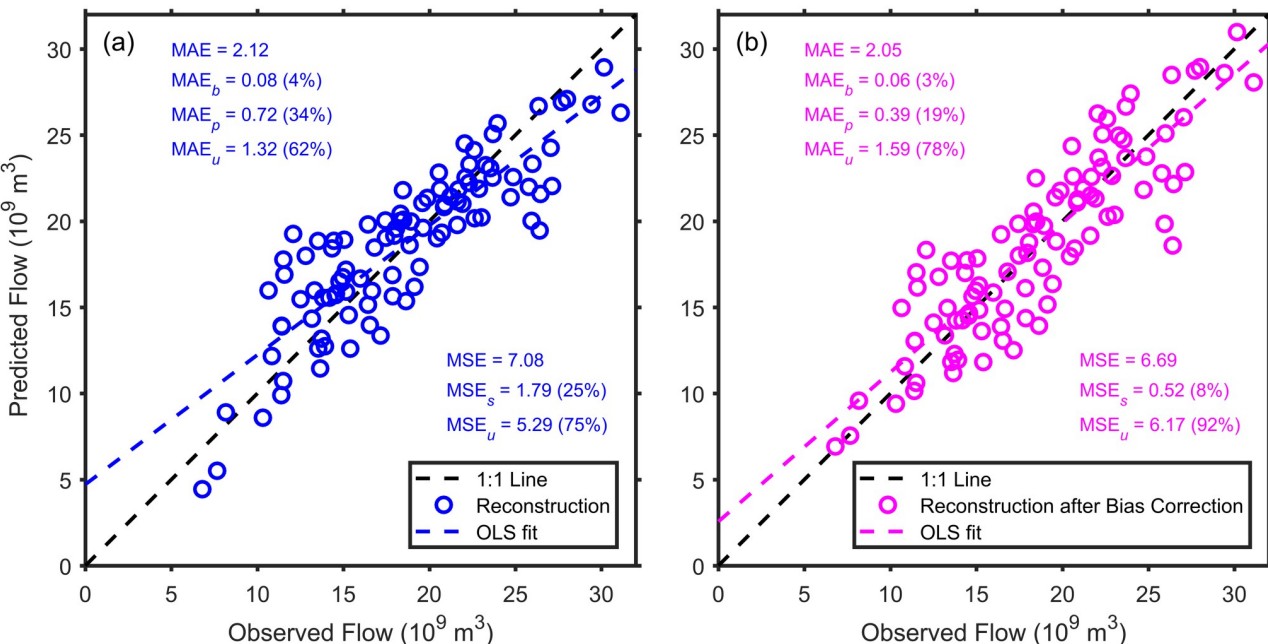

**Fig 2. Model-estimation errors before and after bias correction.** Model-estimation errors for (a) the reconstruction of annual Upper Colorado River flow (in billions of cubic meters) from [15] and (b) the same reconstruction after applying the bias-correction procedure of [14].

model errors to compare the three sources of error in the original and bias-corrected reconstruction.

Prior to bias correction (Fig 2a), the Upper Colorado River reconstruction has low overall error, with a MAE of 2.12 billion m$^3$ (i.e., when compared to the $\bar{O}$ of 18.53 billion m$^3$). The small value of $MAE_b$ (0.08 billion m$^3$) also shows that the reconstruction model faithfully reproduces the observed mean. From the scatterplot and the substantial amount (34%) of error in $MAE_p$, however, it is clear that high flow years are underestimated (and, to a lesser extent, low flows are overestimated). Even with these substantial proportionality errors, the majority (62%) of the mean absolute error is in $MAE_u$), which is desirable (i.e., the majority of error is unsystematic). At the same time, the traditional decomposition into $MSE_s$ and $MSE_u$ masks the distinction between bias and proportionality error while also providing an underestimate of these combined systematic errors because it is inflating its representation of the unsystematic error ($MSE_u$) by squaring the model-predicted deviations from the regression line. As a result, the MSE-based measures suggest that there is little room for improvement when there is.

Bias correction (Fig 2b) produces a reconstruction model that has similar MAE (2.05 billion m$^3$) to the original model. But, bias correction has produced much lower error in the two systematic terms of MAE, reducing $MAE_b$ to 3% and $MAE_p$ to 19% of MAE. From the slope of the regression line, however, it is clear that there still is some proportionality error that the bias-correction procedure has not entirely removed. The MSE-based measures present a rosier picture of the reduction of systematic error, again due to the inflation of the unsystematic error produced by the squaring of the deviations around the regression line. Overall, the MAE-based approach shows that there is still room for additional improvement in the original reconstruction (Fig 2a) and in the bias correction procedure (Fig 2b) than is evident in the MSE-based measures. In particular, the additional systematic component introduced here, $MAE_p$, suggests that high flows still need to be adjusted upward.

## Conclusions

Traditional decomposition of sums-of-squared errors into systematic ($MSE_s$) and unsystematic ($MSE_u$) components has been a popular approach for characterizing the different components of model error. These sums-of-squares-based measures, however, have been shown to be imprecise and, at times, deceptive indicators of average error and its constituents. As a result, evaluations of model estimates and predictions should increasingly use absolute-value-based error measures such as MAE. To fill the need for a decomposition of MAE into its constituent components, we present new measures that are formed as weighted averages of the absolute error. As a result, MAE can now be decomposed into three components that represent bias ($MAE_b$), proportionality ($MAE_p$), and unsystematic ($MSE_u$) errors. These measures provide a more intrepretable standard for evaluating model errors while also pointing to more specific types of error that may be reduced.

## Supporting information

**S1 File.**
(PDF)

## Acknowledgments

The authors appreciate the informative and constructive comments of the two reviewers.

## Author Contributions

**Conceptualization:** Scott M. Robeson, Cort J. Willmott.

**Formal analysis:** Scott M. Robeson.

**Methodology:** Scott M. Robeson, Cort J. Willmott.

**Project administration:** Scott M. Robeson.

**Visualization:** Scott M. Robeson.

**Writing – original draft:** Scott M. Robeson.

**Writing – review & editing:** Scott M. Robeson, Cort J. Willmott.

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
