## [Decision Letter · Decision Letter 0]

26 Oct 2022

PONE-D-22-17609

Decomposition of the mean absolute error (MAE) into systematic and unsystematic components

PLOS ONE

Dear Dr. Robeson,

Thank you for submitting your manuscript to PLOS ONE. After careful consideration, we feel that it has merit but does not fully meet PLOS ONE’s publication criteria as it currently stands. Therefore, we invite you to submit a revised version of the manuscript that addresses the points raised during the review process.

We look forward to receiving your revised manuscript.

Kind regards,

Fabiana Zama

Academic Editor

PLOS ONE

Journal Requirements:

Additional Editor Comments:

Dear Prof. Scott M. Robeson,

Based on the reviewers' comments, the decision is to accept your manuscript after minor revisions.

Please follow the points raised by each reviewer.

Kind Regards

Fabiana Zama

Reviewers' comments:

Reviewer's Responses to Questions

**Comments to the Author**

1. Is the manuscript technically sound, and do the data support the conclusions?

Reviewer #1: Yes

Reviewer #2: Yes

2. Has the statistical analysis been performed appropriately and rigorously? 

Reviewer #1: Yes

Reviewer #2: Yes

3. Have the authors made all data underlying the findings in their manuscript fully available?

Reviewer #1: No

Reviewer #2: Yes

4. Is the manuscript presented in an intelligible fashion and written in standard English?

Reviewer #1: Yes

Reviewer #2: Yes

5. Review Comments to the Author

Reviewer #1: My areas of expertise focus on metrics to measure error and change, which is the exact topic of the submitted manuscript. I have thought about the concepts of the submitted manuscript for decades. Therefore, I report with confidence that the submitted manuscript is a brilliant major breakthrough. I inserted several comments in the PDF as I read. I ask the authors to consider those comments to increase clarity of the manuscript. The submitted manuscript is short, which is a strength. The manuscript states only what is necessary, which any scientific manuscript should do. The manuscript proposes a method to fix the flawed popular paradigm of squared deviations. The authors’ new method makes much better sense than the popular paradigm. Previous methods that have used Mean Absolute Error (Pontius Jr 2022) do not separate the Mean Absolute Error into as many helpful components as the proposed manuscript does. The manuscript illustrates how the new method has helpful practical implications. Below are ideas to make the manuscript even stronger than it already is.

My browser could not activate the link where the authors have posted the data at https://www.ncdc.noaa.gov/dataaccess/paleoclimatology%E2%80%90data

The example is helpful. It would be more helpful to have a column at the right in Table 1 to show the sum. It would be nicer to have the numbers be simpler such as all whole numbers for Pi and Oi, and to have n be a number that makes easier division than by 7.

In figure 1d, it is not immediately clear to me why MAEb = 0.424 rather than |-0.714|, which is the absolute bias in figure 1c. The reader would understand better if the revised manuscript were to have a sentence to explain why |Bias| does not equal MAEb.

Figures 1 and 2 should be consistent in the number of digits in the results. Report the % to the nearest whole number. The other numbers should have exactly two decimal places.

I thank the authors for using sequential line numbers.

In line 108, I think it would be clearer to eliminate “are conservative and must”

In line 137, the meaning of the quotes around average is unclear. I find the language is imprecise when I see quotes like that.

In line 155, replace “powerful” with “popular”.

In lines 139-140, the implication is profound. Congratulations on the creation of a helpful method.

Readers will be eager to have computer code, say in R, which would inspire more rapid adoption by researchers.

I hope the authors find this review helpful, as I intend it to be. The authors have achieved a major accomplishment.

LITERATURE

Pontius Jr, Robert Gilmore. 2022. Metrics That Make a Difference: How to Analyze Change and Error. Advances in Geographic Information Science. Cham: Springer International Publishing. https://doi.org/10.1007/978-3-030-70765-1.

Reviewer #2: In the present paper, the authors propose the decomposition of the Mean Absolute Error into three components which represent the model bias, proportionality, and unsystematic components.

The error components are clearly explained through a synthetic example and a data sample.

Therefore, the present study improves the standard measures for evaluating model errors significantly. The analysis of the three components makes it possible to understand their different contribution.

Overall, the paper is well written and well organized, hence the decision is to accept after minor revisions.

Points of attention:

• The data link is not working: https://www.ncdc.noaa.gov/dataaccess/ paleoclimatology‐data

• Page 2 line 16. remove parenthesis

• Page 2 line 41. remove parenthesis

• Page 4 equation (14-16). Should be divided by n.

• Page 5 equation (17) holds due to the definitions in (14)-(16).

6. PLOS authors have the option to publish the peer review history of their article (what does this mean?). If published, this will include your full peer review and any attached files.

Reviewer #1: **Yes: **Robert Gilmore Pontius Jr

Reviewer #2: No

---

## [Author Response · Author response to Decision Letter 0]

9 Dec 2022

Decomposition of the mean absolute error (MAE) into

systematic and unsystematic components

Response to Reviews

Reviewer #1 

My areas of expertise focus on metrics to measure error and change, which is the exact topic of the submitted manuscript. I have thought about the concepts of the submitted manuscript for decades. Therefore, I report with confidence that the submitted manuscript is a brilliant major breakthrough. I inserted several comments in the PDF as I read. I ask the authors to consider those comments to increase clarity of the manuscript. The submitted manuscript is short, which is a strength. The manuscript states only what is necessary, which any scientific manuscript should do. The manuscript proposes a method to fix the flawed popular paradigm of squared deviations. The authors’ new method makes much better sense than the popular paradigm. Previous methods that have used Mean Absolute Error (Pontius Jr 2022) do not separate the Mean Absolute Error into as many helpful components as the proposed manuscript does. The manuscript illustrates how the new method has helpful practical implications. Below are ideas to make the manuscript even stronger than it already is.

Thank you so much for your comments and the very useful feedback throughout your review. 

My browser could not activate the link where the authors have posted the data at https://www.ncdc.noaa.gov/dataaccess/paleoclimatology%E2%80%90data

Our apologies. We have corrected the link, which now goes directly to the NOAA site for the particular data used. It also is given below:

https://www.ncei.noaa.gov/access/paleo-search/study/28810

The example is helpful. It would be more helpful to have a column at the right in Table 1 to show the sum. It would be nicer to have the numbers be simpler such as all whole numbers for Pi and Oi, and to have n be a number that makes easier division than by 7.

We modified our example to use a set of 6 numbers. We also added the summation column at the far right. The figure has been updated accordingly.

In figure 1d, it is not immediately clear to me why MAEb = 0.424 rather than |-0.714|, which is the absolute bias in figure 1c. The reader would understand better if the revised manuscript were to have a sentence to explain why |Bias| does not equal MAEb.

We have added the following explanation at the end of the section that discusses Fig. 1, as well as the additional recommendation to continue examining MBE:

The constraints within the weighted decomposition of MAE diminish MAEb relative to the magnitude of MBE. MBE, therefore, remains a useful metric to be reported when analyzing model error. 

Figures 1 and 2 should be consistent in the number of digits in the results. Report the % to the nearest whole number. The other numbers should have exactly two decimal places.

We made these corrections. 

I thank the authors for using sequential line numbers.

In line 108, I think it would be clearer to eliminate “are conservative and must”

In line 137, the meaning of the quotes around average is unclear. I find the language is imprecise when I see quotes like that.

In line 155, replace “powerful” with “popular”.

We made minor edits to address all of these comments.

In lines 139-140, the implication is profound. Congratulations on the creation of a helpful method.

Thank you!

Readers will be eager to have computer code, say in R, which would inspire more rapid adoption by researchers.

We now provide R and Matlab functions for these calculations in the Supporting Information. 

I hope the authors find this review helpful, as I intend it to be. The authors have achieved a major accomplishment.

We found this review extremely helpful and appreciate your positive assessment. 

LITERATURE

Pontius Jr, Robert Gilmore. 2022. Metrics That Make a Difference: How to Analyze Change and Error. Advances in Geographic Information Science. Cham: Springer International Publishing. https://doi.org/10.1007/978-3-030-70765-1.

This reference was added and we also made the suggested editorial changes that were in the annotated PDF. 

Reviewer #2

In the present paper, the authors propose the decomposition of the Mean Absolute Error into three components which represent the model bias, proportionality, and unsystematic components.

The error components are clearly explained through a synthetic example and a data sample.

Therefore, the present study improves the standard measures for evaluating model errors significantly. The analysis of the three components makes it possible to understand their different contribution.

Overall, the paper is well written and well organized, hence the decision is to accept after minor revisions.

Thank you very much for your comments and for the overall positive assessment of our work. 

Points of attention:

• The data link is not working: https://www.ncdc.noaa.gov/dataaccess/ paleoclimatology‐data

Our apologies. We have added the corrected link, which now goes directly to the NOAA site for the particular data used. It also is given below:

https://www.ncei.noaa.gov/access/paleo-search/study/28810

• Page 2 line 16. remove parenthesis

• Page 2 line 41. remove parenthesis

We made these two corrections.

• Page 4 equation (14-16). Should be divided by n.

Thank you for catching this error. It has been corrected.

• Page 5 equation (17) holds due to the definitions in (14)-(16).

We changed the text here slightly to clarify this point.

---

## [Editor Report · Decision Letter 1]

14 Dec 2022

Decomposition of the mean absolute error (MAE) into systematic and unsystematic components

PONE-D-22-17609R1

Dear Dr. Robeson,

We’re pleased to inform you that your manuscript has been judged scientifically suitable for publication and will be formally accepted for publication once it meets all outstanding technical requirements.

Kind regards,

Fabiana Zama

Academic Editor

PLOS ONE
---

## [Editor Report · Acceptance letter]

21 Dec 2022

PONE-D-22-17609R1 

Decomposition of the mean absolute error (MAE) into systematic and unsystematic components 

Dear Dr. Robeson:

I'm pleased to inform you that your manuscript has been deemed suitable for publication in PLOS ONE. Congratulations! Your manuscript is now with our production department. 

Kind regards, 

on behalf of

Professor Fabiana Zama 

Academic Editor

PLOS ONE